Predictors of preterm births in North Dakota: a retrospective study of the North Dakota Pregnancy Risk Assessment Monitoring System (PRAMS)

Njau Grace 1 2
Danielson Ramona 3
Day Corey 2
Odoi Agricola aodoi@utk.edu 2
1 Division of Special Projects & Health Analytics, North Dakota Department of Health and Human Services , Bismarck , ND , United States of America
2 Biomedical and Diagnostic Sciences, University of Tennessee , Knoxville , TN , United States of America
3 Department of Public Health, North Dakota State University , Fargo , ND , United States of America
Adegboye Oyelola
Electronic publication date: 2025 Mar 17
Publication date: 2025
Volume: 13
Electronic Location ID: e19049
Received 2024 Jun 19; Accepted 2025 Feb 3
Copyright: ©2025 Njau et al.
Copyright year: 2025
Copyright holder: Njau et al.
License: This is an open access article distributed under the terms of the Creative Commons Attribution License, which permits unrestricted use, distribution, reproduction and adaptation in any medium and for any purpose provided that it is properly attributed. For attribution, the original author(s), title, publication source (PeerJ) and either DOI or URL of the article must be cited.
License URL: https://creativecommons.org/licenses/by/4.0/

Keywords: Preterm birth, Pregnancy Risk Assessment Monitoring, PRAMS, Logistic regression, North Dakota, United States

Funding: The authors received no funding for this work.

==============================
Background

Preterm births represent approximately 10% of all births in the United States (US) annually. Although North Dakota (ND) has large rural and American Indian populations that experience disparities in health outcomes relative to the general population, few studies have investigated risk factors of preterm births in this state. Therefore, the objective of this study was to investigate predictors of preterm births in ND among women who had a live singleton birth and no prior history of preterm births.

Methods

Data on live births from 2017 to 2021 were obtained from the ND Pregnancy Risk Assessment Monitoring System. Potential predictors of preterm birth were identified using a conceptual model. Multivariable logistic regression was then used to investigate and identify significant predictors of preterm births.

Results

The overall prevalence of preterm birth in North Dakota from 2017 to 2021 was 8.1%. However, among the population of interest in this study, which excluded births with multiple infants as well as women with a prior history of preterm birth, the preterm birth rate was 6.4%. Race, age, pregestational diabetes, and gestational hypertension were significantly associated with preterm birth in the final multivariable logistic model. The odds of preterm birth were higher among women who were American Indian (adjusted odds ratio (AOR) = 1.7, 95% confidence interval (CI) [1.3–2.4]), were aged 35 years or older (AOR = 1.6, 95% CI [1.01–2.5]), had pregestational diabetes (AOR = 4.3, 95% CI [2.0–9.3]), and had gestational hypertension (AOR = 4.5, 95% CI [3.1–6.7]) compared to women who were White, aged 20–34 years, and did not have pregestational diabetes or gestational hypertension.

Conclusions

Preventing and controlling chronic diabetes and hypertensive disorders of pregnancy is critical for reducing the risk of preterm birth, especially among women of advanced maternal age. Further research is needed to understand the underlying causes of racial disparities of preterm birth in ND.

Background

Preterm birth (i.e., birth prior to 37 weeks of gestation) is among the top five causes of infant mortality in the United States (US) and causes more than 50% of the global neonatal deaths (Ananth, Ananth & Vintzileos, 2006; Blencowe et al., 2012; Alijahan et al., 2014; Centers for Disease Control and Prevention, 2023a). Premature infants are at increased risk of severe neonatal and infant disorders such as chronic lung disease, metabolic disorders, and developmental delays. In 2022, nearly one in 10 infants in the US were born premature (Martin, Hamilton & Osterman, 2023), and the incidence of premature singleton births in the US increased by 0.7% from 2014 to 2022 (Martin, Hamilton & Osterman, 2023). Southern states, including Mississippi, Louisiana, and West Virginia, are among those with the highest rates of preterm births in the US, with 13% or more of the births being preterm in each state in 2022 (Centers for Disease Control and Prevention, National Center for Health Statistics, 2022). Preterm birth rates are lowest in New Hampshire, Oregon, Washington, Vermont, and Idaho, where fewer than 9% of the births were preterm in 2022 (Centers for Disease Control and Prevention, National Center for Health Statistics, 2022).

In 2005, the total economic toll of preterm births in the US was estimated at $26.2 billion (2005 USD) (Behrman & Butler, 2007), and in 2016, one cohort study found that the individual healthcare costs of infants born preterm was $76,153 (2016 USD) (Beam et al., 2020). The costs associated with preterm births are largely related to maternal and neonatal medical and healthcare costs, early intervention services for the children born premature, special education services, and loss of work and pay for individuals who give birth prematurely (Ananth, Ananth & Vintzileos, 2006; Gyamfi-Bannerman et al., 2011). The combined health and social costs of preterm births warrant urgent attention to identify modifiable risk factors and implement programs to reduce risk in the US and globally.

There are several known risk factors for preterm births. Women who have previously had a preterm birth are substantially more likely to have a preterm birth in subsequent pregnancies (Ekwo, Gosselink & Moawad, 1992). Additionally, multiple gestation (i.e., twins or triplets) is likely to result in preterm birth (Murray et al., 2018). Multiple medical conditions have been associated with preterm births including high blood pressure, underweight and obesity, and diabetes (Berger et al., 2020). Demographic and socioeconomic factors can also be predictive of preterm births. For example, in the US, preterm birth prevalence is significantly higher among non-Hispanic Black and American Indian/Alaska Native women than among non-Hispanic White women (Culhane & Goldenberg, 2011; Hamilton et al., 2019; Abrahamowicz et al., 2023), and lower socioeconomic metrics such as lower income and education are associated with higher risk of preterm birth (Oftedal et al., 2016; Brink et al., 2020). Healthcare access and utilization are additional risk factors, as the odds of preterm births increase with inadequate prenatal care (El-Sayed & Galea, 2012; Zhang et al., 2017). The role of adverse childhood experiences (ACEs) in preterm birth is an emerging area of study, as several recent studies have reported significant relationships between ACEs and preterm births (Sulaiman et al., 2021).

North Dakota (ND) had the 25th lowest rate of preterm births in the US in 2022. The rate increased from 8.8% in 2017 to 9.6% in 2021 and 10.3% in 2022 (Centers for Disease Control and Prevention, National Center for Health Statistics, 2022). Unfortunately, efforts to investigate disparities and risk factors for preterm births in ND have been scarce. There is evidence that the risk of preterm birth is significantly higher among American Indian women compared to White women in ND (Danielson et al., 2018). A study of the risk of preterm births of maternal populations in ND and South Dakota found a positive interaction between maternal ACEs and being unmarried (Testa & Jackson, 2021). However, no studies have been conducted that used multivariable models to identify predictors of preterm births in ND while controlling for the effects of covariates. Consequently, knowledge of predictors of preterm births is limited. However, considering that the strongest predictors of preterm births are multiple-gestation pregnancies and prior history of preterm birth (Ekwo, Gosselink & Moawad, 1992; Murray et al., 2018), this study was limited to predictors of preterm birth among ND women without these two characteristics. Therefore, the aim of this study was to investigate predictors of preterm births among ND women who had live singleton births and no prior history of preterm birth. The study’s hypothesis is that there will be significant associations between the odds of preterm births and socioeconomic, demographic, behavioral, and maternal characteristics as well as access to healthcare.

Materials and Methods

Ethical statement

This study used data from the Pregnancy Risk Assessment Monitoring Survey (PRAMS), which is a population-based survey developed by the Centers for Disease Control and Prevention (CDC) and conducted in partnership with the North Dakota Department of Health and Human Services (ND HHS) (Centers for Disease Control and Prevention, 2023c). The CDC’s protocol has been approved by the CDC’s IRB (Shulman et al., 2018). The protocol of this study was reviewed and approved by the University of Tennessee Institutional Review Board (IRB Number: 21-06599-XM). The need for consent was waived by the IRB because the study utilizes secondary data. Data were provided by the ND HHS through a special request. All identifying pieces of information were removed from the dataset before it was provided to investigators. Privacy of data were maintained by ensuring only study personnel who signed data sharing agreement with ND HHS had access to the data, storing the data in password protected computers only accessible to study investigors and ensuring that all study results are reported in aggregated format only.

Study area, data source, and study population

Portions of this text were previously published as part of a PhD dissertation (Njau, 2022). North Dakota has a population of approximately 773,000 people (United States Census Bureau, 2022). The racial/ethnic distribution of the state is 83% White, 5% American Indian, 3% Black, 2% Asian, 2% are two or more races and 5% are Hispanic or Latino. Women of reproductive age (i.e., 15–44 years) comprise approximately 18% of the population (United States Census Bureau, 2022).

Study data were obtained from the ND PRAMS Program. ND PRAMS utilizes standard CDC PRAMS data collection methodology, which has been described in detail elsewhere (Shulman et al., 2018). A stratified random sample of women identified in the state’s birth certificate dataset is invited to participate in the survey (Shulman et al., 2018). In ND, approximately 14% of women who had live births were sampled between 2017 and 2021. The ND PRAMS response rates for this period were 70.2% (2017), 59.9% (2018), 59.1% (2019), 60.5% (2020), and 59.2 (2021). These response rates exceeded the CDC’s threshold of at least a 50% response rate required for a weighted statewide representative sample (Shulman et al., 2018; Centers for Disease Control and Prevention, 2023c). The CDC accounts for three components in the complex weighting design (sampling weight) calculated for each state and year of data, allowing for generalizability of results to the entire population of the jurisdiction, an adjustment for nonresponse, and an adjustment for noncoverage (Shulman et al., 2018; Centers for Disease Control and Prevention, 2023c).

The PRAMS questionnaire covers an array of topics on maternal behaviors and experiences. It includes questions on smoking, substance abuse, insurance status, prenatal care, other healthcare provider access, and healthcare outcomes, such as preterm births, maternal diabetes, hypertension, and behavioral health. States participating in PRAMS may add questions to the core survey that reflect emerging or state-relevant needs. Due to interest among stakeholders in the state, ND added the adverse childhood experiences (ACEs) module of questions in 2017 (Anda et al., 1999; North Dakota Department of Health and Human Services, 2024).

Mothers with prior preterm births (4.3% of overall dataset) or who had plural births (1.4% of overall dataset) were excluded because preterm birth incidence is known to be excessively high under those conditions (Ekwo, Gosselink & Moawad, 1992; Murray et al., 2018). These criteria resulted in the exclusion of 5.7% of live births from the original dataset.

Descriptive analyses

Statistical analyses were performed in SAS software version 9.4 (SAS Institute Inc., 2013) using procedures for complex survey designs with Taylor Series variance estimation. Sampling strata and sampling weight were specified in all analyses. For descriptive analyses, the ‘proc surveyfreq’ procedure was used to calculate weighted frequencies for each variable and the prevalence of preterm births corresponding to each variable (SAS Institute Inc., 2016a).

Investigation of predictors of preterm births

Preterm birth was defined as a live birth which occurred before 37 weeks of gestation. In accordance with a standard definition of preterm birth that allows for comparison across states and over time, a dichotomous preterm birth variable was created by categorizing gestational period into births prior to 37 weeks (preterm) and births 37 weeks or later (term) (Quinn et al., 2016). While preterm births may be further categorized by severity according to the number of weeks of gestation, severity of prematurity was not the focus of the present study. More recent recommendations are to classify preterm birth using a more robust, multi-faceted taxonomy that considers maternal placental factors as well as fetal conditions and signs indicating the beginning of labor, and which are then postnatally confirmed or refined (Villar et al., 2024). This classification goes beyond the scope of this study, as variables needed for the taxonomy were not available in the dataset provided by ND HHS and therefore postnatal confirmation was not possible.

Potential predictors were identified using a conceptual causal model based on existing knowledge (Fig. 1). The predictors consisted of demographic characteristics (i.e., maternal race, maternal age, marital status, residence), behavioral factors (i.e., drinking, smoking cigarettes, and using e-cigarettes three months prior to pregnancy, using marijuana one month prior to pregnancy), maternal characteristics (i.e., adverse childhood experiences (ACE) score, intendedness of the pregnancy, experience of physical abuse during pregnancy), and healthcare access (i.e., whether prenatal care visits were paid by Medicaid, adequacy of prenatal care). The predictors also included pregestational health conditions (i.e., pregestational diabetes, pregestational hypertension, and pregestational body mass index (BMI)) and gestational health conditions (i.e., gestational diabetes, gestational hypertension, depression during pregnancy). While fertility treatments are considered a risk factor for preterm birth (Salmeri et al., 2024; Sanders et al., 2022), the unweighted count of births to women in our original dataset (i.e., prior to removing multiple births and births to women with a prior preterm birth) and in our subset of data from 2017 to 2021 who identified as using assisted reproductive technology or using fertility enhancing drugs were each fewer than 10. Due to the small numbers, fertility treatment was not assessed as a potential predictor in the analyses.

Figure 1 Conceptual causal model of preterm birth among North Dakota women who had live singleton births and did not have a prior preterm birth, 2017–2021.

Potential predictors of preterm birth were identified using a conceptual causal model based on existing knowledge. The potential predictors identified consisted of demographic characteristics, behavioral factors, maternal characteristics, healthcare access, pregestational health conditions, and gestational health conditions. Maternal race, maternal age, geographic residence, and maternal adverse childhood experiences (ACE) score were considered as important potential confounders in the conceptual model.

Since the outcome (preterm births) is dichotomous, logistic regression was used to assess associations between preterm births and several potential predictors listed above and to compute adjusted odds ratios (Sperandei, 2014). Binary logistic regression models were fitted using the ‘proc surveyreg’ procedure of SAS (SAS Institute Inc., 2016b). Model building followed a two-step process (Fig. 2). First, bivariate associations of each potential predictor and preterm birth were assessed by fitting univariable logistic regression models of preterm births for each predictor. Second, predictors were retained if the p-value of a likelihood ratio test comparing the univariable model to the null model was <0.2 (Dohoo, Martin & Stryhn, 2012) and then assessed in a full binary multivariable logistic regression model. A final reduced, parsimonious multivariable model was produced using a manual backward elimination process in which covariates with the highest p-values were removed, one at a time, until all remaining variables were significantly associated with the outcome at a significance threshold p < 0.05. At each elimination step, the coefficients of remaining predictors before and after the removal of each covariate were compared to identify confounders. Non-significant variables were retained as confounders if their removal resulted in >20% change in the coefficient of another variable in the model (Dohoo, Martin & Stryhn, 2012). Age and race were retained in the multivariable model regardless of their statistical significance because they were expected to be important confounders in the conceptual model (Fig. 1). ACE score was not retained as a confounder because it did not have a significant association with preterm birth. The models were refitted using the R package ‘survey’ version 4.2.1 to calculate Akaike Information Criterion (AIC) with consideration of complex survey design (Lumley & Scott, 2015; Lumley, 2004).

Figure 2 Flowchart depicting study setup and data processing pipeline of the study of predictors of preterm births in North Dakota.

Results

Descriptive analysis

The weighted survey responses represented 47,195 singleton births among ND women who never had a previous preterm birth from 2017 to 2021. Stratified population frequencies and preterm birth prevalence proportions are presented in Table 1. The study’s target population reflected greater racial diversity than the state’s population overall; 75% of the study population was White, 8% was American Indian, and 17% was other races (Table 1). The average maternal age was 29 years; 13% of the study population was aged 35 years or older, 83% was aged 20–34 years, and 4% was younger than 20 years. Most of the population was married (68%). The population was nearly evenly distributed between counties classified as urban (51%) and rural (49%). The average maternal ACE score was 1.8; 41% of the population had an ACE score of 0, 20% had a score of 1, 11% had a score of 2, 7% had a score of 3, and 21% had an ACE score of 4 or more. Two in five live births (39%) resulted from unintentional pregnancies, 3% of women experienced physical abuse during pregnancy, and 17% experienced depression during pregnancy. A minority of the population smoked three months prior to pregnancy (19%), used electronic cigarettes three months prior to pregnancy (5%), or used marijuana in the month prior to pregnancy (7%), but the majority did consume alcohol 3 months prior to pregnancy (69%). Approximately one in four births (24%) was to a woman who had prenatal care paid for by Medicaid and 1 in five births (22%) was to a woman who received less than adequate prenatal care. More than half of the population was either overweight (28%) or obese (28%) prior to pregnancy, 4% had diabetes prior to pregnancy, and 5% had high blood pressure prior to pregnancy. Approximately 1 in 10 ND women with a live singleton birth and no prior history of preterm birth developed diabetes during pregnancy (9%) and more than 1 in 10 developed hypertension during pregnancy (12%).

Table 1 Demographic and health status characteristics of North Dakota women who had a singleton birth and did not have a prior preterm birth, 2017–2021.

Variable	Weighted frequency (standard error)	Prevalence of preterm birth (%)	95% confidence level	
Race				
White	34,414 (74)	6.5	5.2, 7.7	
American Indian	3,708 (8)	10.6	8.9, 12.4	
Other race/ethnicity	8,073 (16)	4.2	2.2, 6.2	
Age				
<20	1,700 (3)	4.1	1.3, 7.0	
20–34	39, 381 (82)	6.1	5.0, 7.2	
35+	6,114 (12)	9.0	5.9, 12.1	
Marital status				
Not married	15,295 (508)	5.7	4.5, 6.8	
Ever married	31,900 (457)	8.0	6.1, 9.9	
Residence				
Rural	23,052 (528)	6.5	5.1, 8.0	
Urban	24,177 (520)	6.3	4.9, 7.6	
Drinking 3 months before pregnancy				
Yes	14,627 (471)	6.7	5.0, 8.5	
No	23,058 (498)	6.2	5.0, 7.4	
Smoking 3 months before pregnancy				
Yes	8,965 (402)	9.3	6.6, 12.0	
No	37,893 (452)	5.7	4.7, 6.8	
E-cig use 3 months before pregnancy				
Yes	2,154 (216)	4.9	4.7, 8.8	
No	44,695 (402)	6.5	5.3, 7.5	
Marijuana use 1 month before pregnancy				
Yes	3,254 (246)	6.5	5.3, 7.7	
No	34,025 (381)	6.4	3.0, 9.9	
Adverse childhood experiences score				
0	18,441 (39)	5.9	4.4, 7.4	
1	9,111 (19)	6.3	4.0, 8.7	
2	4,847 (10)	7.4	4.0, 10.7	
3	3,267 (6)	5.5	2.5, 8.6	
4+	9,278 (19)	7.7	5.2, 10.2	
Intentional pregnancy				
Yes	28,660 (59)	5.7	4.5, 8.2	
No	18,076 (37)	7.6	5.9, 9.4	
Physical abuse during pregnancy				
Yes	1,126 (140)	8.6	3.0, 14.3	
No	45,969 (380)	6.3	5.3, 7.4	
Insurance paid by Medicaid				
Yes	10,848 (446)	8.2	5.9, 10.6	
No	34,855 (451)	5.7	5.6, 6.8	
Prenatal care				
Inadequate	5,604 (308)	7.8	5.0, 10.6	
Intermediate	4,507 (295)	3.7	1.1, 6.3	
Adequate	35,460 (472)	6.6	5.4, 7.7	
Pregestational diabetes				
Yes	1,656 (179)	18.1	10.2, 26.0	
No	45,137 (399)	6.0	5.0, 7.0	
Pregestational hypertension				
Yes	2,242 (204)	13.0	6.9, 19.0	
No	44,525 (407)	6.0	5.0, 7.0	
Pregestational body mass index				
Underweight	1,038 (148)	2.8	0.0, 6.1	
Normal	19,152 (505)	6.0	4.5, 7.5	
Overweight	12,618 (443)	5.2	3.5, 6.9	
Obese	13,021 (457)	8.1	6.0, 10.2	
Gestational diabetes				
Yes	4,195 (288)	8.7	4.7, 12.6	
No	42,675 (429)	6.2	5.2, 7.2	
Gestational hypertension				
Yes	5,494 (320)	17.5	13.0, 22.2	
No	41,404 (450)	4.9	4.0, 5.9	
Depression during pregnancy				
Yes	8,214 (393)	6.9	4.4, 9.4	
No	38,775 (463)	6.2	5.2, 7.3	

The overall prevalence of preterm births was 6.4% (95% confidence interval (CI) [5.4–7.4]), but there were clear differences in preterm birth prevalence when stratified by some potential predictors (Table 1). Notably, prevalence was substantially higher among women who were American Indian (10.6%), those aged 35 years or older (9.0%), those who smoked prior to pregnancy (9.3%), those with pregestational diabetes (18.1%) or pregestational hypertension (13.0%), and those who developed hypertension during pregnancy (17.5%), compared to the overall study population.

Predictor investigation of preterm births

Variables that had unadjusted associations with preterm births included maternal race, maternal age, intentionality of pregnancy, smoking three months before pregnancy, pregestational BMI, pregestational diabetes, gestational diabetes, pregestational hypertension, gestational hypertension, maternal marital status, prenatal care paid by Medicaid, and adequacy of prenatal care (Table 2). The results of the final multivariable model show that the significant predictors of preterm births in this study population were maternal race, maternal age, pregestational diabetes, and gestational hypertension (Table 3). Pregestational hypertension (confounder of pregestational diabetes) and pregestational BMI (confounder of race and age) were retained as confounders in the final model although they were not significantly associated with the outcome. The odds of preterm births were higher among women who were American Indian (adjusted odds ratio (AOR) = 1.7, 95% CI [1.3–2.3]) compared to those who were White and among women who were aged 35 years or older (AOR = 1.6, 95% CI [1.0–2.5]) compared to those aged 20–34. Additionally, the odds of preterm births were higher among women who had pregestational diabetes (AOR = 4.3, 95% CI [2.0–9.3]) and who developed gestational hypertension (AOR = 4.5, 95% CI [3.1–6.7]) than those who did not have pregestational diabetes or develop gestational hypertension, respectively. The AIC of the final multivariable model was 0.66 (95% CI [0.64–0.68]).

Discussion

This study investigated predictors of preterm births in ND from 2017 to 2021. The findings are consistent with prior research in other populations, with the results demonstrating that pre-pregnancy diabetes and gestational hypertension are strongly associated with high odds of preterm birth. Advanced maternal age and race have associations that are likely partially or fully mediated by other covariates in the study.

Table 2 Univariable binary logistic regression model results showing variables assessed for simple (unadjusted) associations with preterm birth among North Dakota women who had a singleton birth and did not have a prior preterm birth, 2017–2021.

Variable	Odds ratio	95% confidence interval	p-value*	
Race			0.001	
American Indian	1.7	1.3, 2.3		
Other race/ethnicity	0.6	0.4, 1.1		
White (reference)				
Age			0.04	
<20	0.7	0.3, 1.4		
35+	1.5	1.0, 2.3		
20–34 (reference)				
Geographic residence			0.7	
Rural	0.9	0.7, 1.3		
Urban (reference)				
Adverse childhood experiences score			0.5	
1	1.1	0.7, 1.7		
2	1.3	0.7, 2.2		
3	0.9	0.5, 1.8		
4+	1.3	0.8, 2.1		
0 (Reference)				
Intentional pregnancy			0.02	
Yes	0.7	0.5, 1.0		
No (reference)				
Physical abuse during pregnancy			0.39	
Yes	1.4	0.7, 2.9		
No (reference)				
Depression during pregnancy			0.6	
Yes	1.1	0.7, 1.7		
No (reference)				
Drinking 3 months before pregnancy			0.5	
Yes	0.9	0.6, 1.3		
No (reference)				
Smoking 3 months before pregnancy			0.001	
Yes	1.7	1.2, 2.4		
No (reference)				
E-cig use 3 months before pregnancy			0.4	
Yes	0.7	0.3, 1.8		
No (reference)				
Marijuana use 3 months before or during pregnancy			0.4	
Yes	0.7	0.3, 1.8		
No (reference)				
Pregestational body mass index			0.03	
Underweight	0.5	0.1, 1.6		
Overweight	0.9	0.5, 1.4		
Obese	1.4	0.9, 2.1		
Normal (reference)				
Pregestational diabetes			<0.001	
Yes	3.4	2.0, 6.0		
No (reference)				
Gestational diabetes			0.1	
Yes	1.4	0.9, 2.4		
No (reference)				
Pregestational hypertension			0.001	
Yes	2.3	1.3, 4.1		
No (reference)				
Gestational hypertension			<0.001	
Yes	4.1	2.8, 6.0		
No (reference)				
Marital status			0.008	
Married	1.5	1.0, 2.0		
Not married (reference)				
Prenatal care paid by Medicaid			0.01	
Yes	1.5	1.0, 2.1		
No (reference)				
Prenatal care			0.05	
Inadequate	1.2	0.8, 1.9		
Intermediate	0.6	0.3, 1.2		
Adequate (reference)				
Notes.

* Statistical significance for the univariable analysis was assessed using a likelihood ratio test critical p = 0.20.

Pregestational diabetes was a significant predictor of preterm births in this study, which confirmed findings of at least one previous study (Sibai et al., 2000). The control of pregestational diabetes during pregnancy is challenging and requires specialized care, including continuous re-assessment of insulin requirements throughout pregnancy (American College of Obstetricians and Gynecologists, 2018; American Diabetes Association, 2020; Hart, Shubrook & Mason, 2021). Approximately 11.6% of the US population is estimated to have diabetes, with most diabetes occurring in people over the age of 44 years (Centers for Disease Control and Prevention, 2023b). However, the burden of diabetes is expected to increase substantially among US youth in the near future (Tönnies et al., 2022), which will likely result in more pregnancies being affected by pregestational diabetes. Importantly, a recent study found that among US women of reproductive age who had diabetes, nearly 30% had undiagnosed diabetes and more than 50% had uncontrolled diabetes (Azeez et al., 2019).

The results of this study are consistent with existing evidence that hypertension during pregnancy is an important risk factor for preterm births (Macdonald-Wallis et al., 2014; Premkumar et al., 2019; An et al., 2022). Among the risk factors of gestational hypertension are obesity, pregestational diabetes and hypertension, and advanced maternal age (The American College of Obstetricians and Gynecologists, 2013). One cohort study of singleton births in Canada revealed that the risk of preterm births among pregnancies with hypertensive disorders was four times higher than normotensive pregnancies, and, notably, women with pre-pregnancy hypertension which progressed to pre-eclampsia had a nearly 45-fold increase in risk of preterm births (Berger et al., 2020). Gestational hypertension is also a strong risk factor of post-pregnancy cardiovascular disease (Benschop, Duvekot & Roeters van Lennep, 2019; Dall’Asta et al., 2021).

Table 3 Final multivariable binary logistic regression model results showing predictors of preterm birth among North Dakota women who had a singleton birth and did not have a prior preterm birth, 2017–2021.

Variable	Odds ratio	95% confidence interval	p-value*	
Race			0.002	
American Indian	1.7	1.3, 2.3	0.001	
Other race/ethnicity	0.8	0.5, 1.3	0.4	
White (reference)				
Age (years)			0.01	
<20	0.4	0.2, 1.0	0.049	
35+	1.6	1.0, 2.5	0.04	
20–34 (reference)				
Pregestational diabetes				
Yes	4.3	2.0, 9.3	<0.001	
No (reference)				
Gestational hypertension				
Yes	4.5	3.1, 6.7	<0.001	
No (reference)				
Pregestational hypertension*				
Yes	0.7	0.3, 1.3	0.4	
No (reference)				
Pregestational body mass indexa			0.2	
Underweight	0.5	0.1, 1.3	0.1	
Overweight	0.7	0.5, 1.1	0.2	
Obese	1.0	0.7, 1.5	0.9	
Normal (reference)				
Notes.

* Statistical significance for the multivariable analysis was assessed using a likelihood ratio test critical p = 0.05.

a Retained as confounders because eliminating them from the model resulted in >20% change in the coefficient of another variable in the model.

The average maternal age in the United States is increasing, resulting in greater risk of adverse pregnancy outcomes including preterm births (Fuchs et al., 2018; Hamilton et al., 2019). In this study, the odds of preterm births increased with age. The apparent linear association between age and preterm birth in this study differs from a 2018 analysis of maternal age and preterm births in Canada, which found a U-shaped distribution where preterm birth rates were higher among women younger than 20 years as well as those aged 35 years or older compared to women aged 20–34 years (Fuchs et al., 2018). Notably, the association between advanced maternal age and preterm births in this study remained significant in the final model even after controlling for BMI, pregestational diabetes, and hypertension that arose before or during pregnancy, which are sometimes attributed as underlying causes of increased rates of adverse pregnancy outcomes among older women (Obstetrics and Gynecology, 2022).

The results of this study are consistent with previous evidence that American Indian women giving birth in ND have significantly higher risk of preterm births than White women (Danielson et al., 2018; Testa & Jackson, 2021). The American Indian population has severe health disparities relative to the US population overall, and there is evidence that American Indian health disparities within the Northern Great Plains region (including ND) are more severe than other regions of the US (Johnson, Call & Blewett, 2010; Indian Health Services, 2017). Notably, other studies have found that racial disparities of preterm births are not entirely mediated by socioeconomic conditions, although relatively few studies have included large American Indian populations (Manuck, 2017).

There was no significant association between maternal ACE score or maternal residence in rural areas with the odds of preterm birth in this study, but other studies have identified those factors as important drivers of adverse maternal health outcomes. As North Dakota is a maternity health desert, it is valuable to consider addressing barriers to prenatal care, through innovations like telehealth (Shmerling et al., 2022). Associations between ACEs and adverse pregnancy outcomes, including preterm births, have been reported in multiple studies (Sulaiman et al., 2021). One prospective study in Connecticut and Massachusetts reported that early trauma was associated with decreased gestational age, lower birthweight, and increased risk of gestational hypertension (Smith, Gotman & Yonkers, 2016). A study that used PRAMS data from ND and South Dakota found that maternal ACEs were not significantly associated with preterm births after adjusting for covariates, but demonstrated a positive interaction of maternal ACEs and lack of paternal involvement among unmarried women (Testa & Jackson, 2021). Traumatic, chronic stressors are known to induce a considerable physiological response based on the body’s fundamental need to maintain homeostasis (Latendresse, 2009; American Academy of Pediatrics, 2014; Smith, Gotman & Yonkers, 2016).

Although ND is a geographically large state with half of its births from 2017 to 2021 occurring among women living in rural areas, urban/rural setting was not a significant predictor of preterm birth in this study. These results may be reflective of the healthcare system in ND and not generalizable to other rural settings. For example, rates of intensive care unit admissions and maternal mortality for US live births from 2016 to 2019 in the US were significantly higher in rural areas than urban areas (Harrington et al., 2023).

Strengths and limitations

This is one of the first studies that has investigated predictors of preterm births in ND. It is also one of a few studies within a growing body of work exploring the association between childhood adversity and poor maternal outcomes. With response rates ranging from 59.1% to 70.2% for the five years of data included in this study, the study’s response rates exceeded the threshold established by the CDC and hence the findings are generalizable to women in ND. The sampling strategy for ND PRAMS also ensured a representative sample of American Indian women. Furthermore, the sampling strategy ensured ample sample size for investigation of urban/rural status as a predictor of preterm births. In limiting our study population specifically to women who did not have a previous preterm birth and who had a singleton live birth, our results provide greater focus regarding the common risk factors for preterm births among women without the strongest risk factors of plural births and prior preterm birth. Another key strength of the study was the utilization of clinically reported data for the outcome of interest (gestational age at delivery) which minimizes recall bias.

A limitation of the study was the use of self-reported data. Some PRAMS elements, including the ACEs module, are self-reported during the postpartum period, hence some self-reporting and/or recall bias is to be expected. One study comparing ACEs information collected prospectively (i.e., during childhood) and ACEs reported retrospectively (i.e., by the same individual as an adult) found that the association between the two calculated scores had modest agreement (Reuben et al., 2016). The prospective ACE score was more strongly associated with poorer objective health outcomes (i.e., collected through tests) among adults and the retrospective ACE score was related to poorer subjective outcomes (i.e., self-reported). The authors emphasize that the two scores should be considered to have assessed complementary sources of information, rather than concluding that the retrospective score is inaccurate. In a study of 2003 PRAMS data among 12 states, validity and reliability for three measures (i.e., participation in the Women, Infants, and Children (WIC) program; Medicaid payment for delivery; and breastfeeding initiation) were high and provide evidence for confidence in self-reported PRAMS information (Ahluwalia, Helms & Morrow, 2013).

Additionally, the birth certificate information and the PRAMS dataset do not encompass all known predictors. For example, clinical-level questions on known mechanical risk factors, such as amniocentesis procedures and placental complications, are not included in the PRAMS survey or birth certificate. Thus, this analysis was limited to the available population-level variables. The number of women that used fertility treatments in this study was not large enough for appropriate assessment of this variable for potential association with preterm births. Additionally, information needed for a more robust classification of preterm birth (Villar et al., 2024) was not available in the dataset used for this study, and the dichotomous variable based solely on gestational age at birth can limit the ability to draw clinical insights. However, the dichotomous definition used in this study is consistent with the definition used in extant literature and allows for comparison across jurisdictions and over time. The above limitations notwithstanding, the findings of this study are important in guiding future studies and public health efforts to address preterm births.

Conclusions

There are important disparities in the risk of preterm births by race and age in ND. The results of this study are useful for the ND HHS and clinicians in ND to identify individuals at high risk of preterm birth and to allocate resources aimed at improving maternal health outcomes. Further research into preterm birth as an outcome using the robust classification system would provide valuable insights for clinical practice. Pregestational diabetes and gestational hypertension were important predictors, further demonstrating the importance of prevention and control of these conditions to improve maternal health outcomes, especially among women who are 35 years or older or who are American Indian. Improving infant and maternal health outcomes will require reducing Type 2 diabetes prevalence among women of reproductive age and ensuring that women who have diabetes are properly diagnosed and achieve adequate control before and during pregnancy. Unfortunately, the management of hypertension using antihypertensive medication among women planning for pregnancy is challenging due to associations between some medications and abnormal fetal development (Sinkey et al., 2020). More research is needed to better understand the effectiveness and safety of different mitigation strategies to reduce hypertension risk during pregnancy. Odds of preterm births increased with age, which is a growing concern as the average maternal age in the US increases. Although it is advised that women of advanced reproductive age should aim to reduce their risk factors for preterm births, including hypertension and diabetes, it is important to recognize that preventing those conditions does not entirely mitigate age-associated risks.

American Indian women have significantly higher risk of preterm births than White women. Further research is warranted to better understand the factors underlying disparities in preterm birth between American Indian and White women in ND. The findings of the current study suggest a lack of association between ACEs and preterm births in this population even without controlling for covariates, but evidence from other studies warrants additional research into the relationships between early trauma and maternal health outcomes. Further research into the relationship between geographic setting and other infant and maternal health outcomes is also warranted. Suffice it to say that the findings of this study provide insight into predictors of preterm births in ND and will be useful for guiding state health programming as well as individual healthcare systems’ efforts to improve maternal and child health in the state.

Supplemental Information

Supplemental Information 1 Codebook

Supplemental Information 2 Raw data

All variables used in the analysis.

Additional Information and Declarations

Competing Interests

Author Contributions

Human Ethics

Data Availability

Agricola Odoi is an Academic Editor for PeerJ.

Grace Njau conceived and designed the experiments, performed the experiments, analyzed the data, prepared figures and/or tables, authored or reviewed drafts of the article, and approved the final draft.

Ramona Danielson performed the experiments, prepared figures and/or tables, authored or reviewed drafts of the article, and approved the final draft.

Corey Day performed the experiments, analyzed the data, prepared figures and/or tables, authored or reviewed drafts of the article, and approved the final draft.

Agricola Odoi conceived and designed the experiments, performed the experiments, authored or reviewed drafts of the article, and approved the final draft.

The following information was supplied relating to ethical approvals (i.e., approving body and any reference numbers):

University of Tennessee Institutional Review Board. IRB Number: 21-06599-XM.

The following information was supplied regarding data availability:

The raw data are available in the Supplemental File.

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
