# Peer review of "Predictors of preterm births in North Dakota: a retrospective study of the North Dakota Pregnancy Risk Assessment Monitoring System (PRAMS)"

_PeerJ, doi:10.7717/peerj.19049_

## Round 0.1 · original submission · Major Revisions

The reviewers have commended the clarity and professionalism of your language and recommended significant revision of your manuscript.

Reviewer 1 ·

Basic reporting

Clear and unambiguous, professional English used throughout. Follows the reporting guideline of the journal.

Experimental design

Lines 144-145 mentioned that the "ND PRAMS response rates were 70.2%, 59.9%, 59.1%, 60.5% and 59.2%" during 2017-2021. The authors should discuss whether there is potential selection bias leading the sample to fail to represent the population.

In the abstract, the authors mentioned two models: one is the full model, and the other one is the reduced model. The authors discussed the results from both models and the model results in discrepancies between the two models, especially regarding the race effect. The authors should use one final model based on a model selection approach, such as the goodness of fit test, to select the best model instead of presenting the results based on multiple models.

Validity of the findings

See my comment on the design.

Reviewer 2 ·

Basic reporting

The presentation is appropriate, references may be expanded, all other reporting components appear appropriate.

Experimental design

Study Object is well suited with the scopes of the Journal , study design and methods are sound. Statistial analysis appears acceptable. I have no further comment on the methods.

Validity of the findings

Internal validity is very likely. I am not sure about external valitidy since this is a peculiar population with lower PTB rate compared to US in general. The study may serve at a local level to derive policies and protocols of individual local hospitals.

Additional comments

I have the following additional and important issues for the authors:

1. PTB is traditionally defined by gestational age at birth (˂37 weeks). However this is a simplistic approach that limits clinical usefulness and hinders progress in identifying effective interventions. A recent consensus paper proposes a functional taxonomy of PTB based on four elements: (1) predefined conceptual principles, (2) known etiologic factors, (3) specific obstetric and neonatal clinical phenotypes identified prospectively, and (4) postnatal growth and development follow-up up to 2 years. This taxonomy incorporates maternal, placental, and fetal conditions routinely recorded in data collection systems. Given the remarkable sample size collected by the authors in their study I invite the authors to consider defining some of these phenotypes within their population and adding some subgoup analyses. If they do not deem this feasible or reevant for their scopes, I still reccoment that they may comment on this issue along with relevant reference in the discussion section.

2. A recent major study designed as overview os systematic review shows that singletons conceived through IVF/ICSI have double the risk of PTB compared to natural conception. This risk is higher with fresh embryo transfers, often due to placental issues. As noted by one of the reviewers data on PTB causes and initiation is limited in the study of the authors and conception method is not presented (please add). Please the authors focus on this issue and add major reference on the topic.

Reviewer 3 ·

Basic reporting

Clarity and Professionalism:
The manuscript is generally written in clear and professional English. However, some sections, such as the description of statistical methods on pages 10-12, could be streamlined for better readability. Additionally, minor grammatical errors are present (e.g., page 3, line 15: "there is evidence that preterm birth risk is significantly higher").

Literature References and Background:
The manuscript provides a thorough review of relevant literature, but it would benefit from the inclusion of more recent studies. For instance, recent advancements in prenatal care and telehealth interventions could be discussed on page 4, lines 40-50. This would provide a more current perspective and highlight the novelty of the approach.

Structure, Figures, and Tables:
The structure of the article conforms to PeerJ standards. Figures and tables are relevant, high quality, well labeled, and described. However, some figures (e.g., Figure 1 on page 11) could benefit from more detailed legends explaining the data shown. Additionally, the captions of tables (e.g., Table 3 on page 14) should be more descriptive to ensure they are understandable without referring to the main text.

Self-contained with Relevant Results:
The manuscript is self-contained, with results that directly address the stated hypotheses. The authors effectively present their findings, which are relevant to their research questions. The inclusion of detailed experimental data and analysis supports the conclusions drawn.

Experimental design

Original Research:
The study presents original primary research that fits within the journal’s scope. The research question is well-defined and addresses a significant gap in the literature, specifically the predictors of preterm births in North Dakota. The rationale for the study is clearly stated on page 2, lines 30-40.

Rigorous Investigation:
The investigation is conducted rigorously, adhering to high technical and ethical standards. Ethical approvals and patient consent are appropriately documented on page 5, lines 125-128. However, more detail on patient data privacy protection would be beneficial.

Methods Description:
The methods are described in sufficient detail to allow for replication, but certain sections could benefit from additional clarification. For example, the rationale behind the choice of specific statistical models (pages 10-12) could be elaborated further. Including a flowchart summarizing the experimental setup and data processing pipeline (perhaps on page 11) would enhance clarity.

Validity of the findings

Replication and Novelty:
The manuscript encourages meaningful replication and contributes novel insights. However, potential challenges in replicating the study should be discussed, such as variations in healthcare access across different regions. This could be added to the discussion section on page 16.

Data Robustness and Statistical Soundness:
The data analysis is robust and statistically sound. The authors use appropriate statistical methods to validate their findings. However, the potential limitations of using self-reported data and the impact of recall bias should be addressed (page 17, lines 350-360). Discussing alternative approaches and their potential impact would strengthen this section.

Conclusions:
The conclusions are well-stated and linked to the original research question. However, more specific recommendations for future research and practical applications would enhance this section. For instance, the authors could discuss how their findings could be integrated into clinical practice in more detail (page 18, lines 370-380).

Additional comments

-The introduction provides a comprehensive background but could benefit from a clearer statement of the study's aims and hypotheses (page 2).
-The discussion section effectively contextualizes the findings but should include a more critical analysis of the limitations and potential biases (page 17).
-The use of logistic regression is appropriate, but the authors should discuss the potential limitations and alternative methods considered (pages 10-12).
-The manuscript would benefit from a more detailed explanation of the ethical considerations related to data privacy (page 5).

---

## Round 0.2 · accepted · Accept

All reviewer's comments have been adequately addressed.

Reviewer 1 ·

Basic reporting

Follows the guidelines.

Experimental design

Follows the guidelines.

Validity of the findings

Follows the guidelines.

Reviewer 2 ·

Basic reporting

No comments

Experimental design

No comment

Validity of the findings

No comment

Additional comments

No comment

Reviewer 3 ·

Basic reporting

The authors have addressed my comments. I have no further suggestions.

Experimental design

The authors have addressed my comments. I have no further suggestions.

Validity of the findings

The authors have addressed my comments. I have no further suggestions.

Additional comments

The authors have addressed my comments. I have no further suggestions.